# Sex Differences in Chronic Thromboembolic Pulmonary Hypertension. Treatment Options over Time in a National Referral Center

**DOI:** 10.3390/jcm10184251

**Published:** 2021-09-19

**Authors:** Alejandro Cruz-Utrilla, María José Cristo-Ropero, Miguel Calderón-Flores, Maite Velázquez, María Jesús López-Gude, Yolanda Revilla Ostolaza, José Luis Pérez Vela, Javier de la Cruz-Bértolo, Héctor Bueno, Fernando Arribas Ynsaurriaga, José María Cortina, Pilar Escribano-Subias

**Affiliations:** 1Pulmonary Hypertension Unit, Department of Cardiology, Hospital Universitario 12 de Octubre, 28041 Madrid, Spain; acruzutrilla@gmail.com; 2Centro de Investigación Biomédica en Red en Enfermedades Cardiovasculares, Instituto de Salud Carlos III (CIBERCV), 28041 Madrid, Spain; maitevel05@gmail.com (M.V.); hector.bueno@salud.madrid.org (H.B.); fernando.arribas@salud.madrid.org (F.A.Y.); 3Department of Cardiology, Hospital Universitario de Torrejón, 28850 Torrejón de Ardoz, Spain; mjcristoropero@gmail.com; 4Department of Cardiology, Hospital Universitario 12 de Octubre, 28041 Madrid, Spain; miguel_cf_94@hotmail.com; 5Interventional Cardiology, Department of Cardiology, Hospital Universitario 12 de Octubre, 28041 Madrid, Spain; 6Department of Cardiac Surgery, Hospital Universitario 12 de Octubre, 28041 Madrid, Spain; mjgude@gmail.com (M.J.L.-G.); josemaria.cortina@salud.madrid.org (J.M.C.); 7Department of Radiology, Hospital Universitario 12 de Octubre, 28041 Madrid, Spain; Yolanda.revilla@salud.madrid.org; 8Department of Intensive Care Medicine, Hospital Universitario 12 de Octubre, 28041 Madrid, Spain; perezvela@yahoo.es; 9Epidemiology & Health Service/Technology Asessment, Instituto de Investigación Sanitaria del Hospital Universitario 12 de Octubre, 28041 Madrid, Spain; jdlcruz@h12o.es; 10Centro Nacional de Investigaciones Cardiovasculares (CNIC), Instituto de Salud Carlos III, 28029 Madrid, Spain

**Keywords:** chronic thromboembolic pulmonary hypertension, women in cardiology, pulmonary hypertension

## Abstract

(1) Background: Clinical presentation, disease distribution, or treatment received may provide insights into the reasons contributing to sex differences in chronic thromboembolic pulmonary hypertension (CTEPH). (2) Methods: We evaluated 453 patients (56% women) between 2007–2019. Data was collected from REHAP (Registro Español de Hipertensión Arterial Pulmonar) registry. Two time periods were selected to evaluate the influence of new treatments over time. (3) Results: Women were older. Baseline functional class was worse, and distance walked shorter in women compared with men. Women had higher pulmonary vascular resistances. Despite this, pulmonary endarterectomy (PEA) was carried out in more men, and women received more frequently pulmonary vasodilators exclusively. The 2014–2019 interval was associated with a better survival only among women. Interestingly, women had a more distal disease during this second period of time. (4) Conclusions: Even though women were older, and received invasive treatments less frequently, mortality was similar in both sexes. The introduction of balloon pulmonary angioplasty and the improvement of pulmonary endarterectomy, especially during the last years, could be associated with a survival benefit among women.

## 1. Introduction

Chronic thromboembolic pulmonary hypertension (CTEPH) is a rare form of precapillary pulmonary hypertension (PH) characterized by the combination of both obstruction of the pulmonary circulation and microvascular dysfunction [1]. In the majority of cases, CTEPH is related to a previous episode of pulmonary embolism (PE) [2] leading to an increase in pulmonary vascular resistance (PVR), and subsequently, the failure of the right ventricle, leading to death if untreated [3]. Treatment of CTEPH has changed significantly during the last decade. Currently, pulmonary endarterectomy (PEA) is still the treatment of choice [4]. However, the introduction of balloon pulmonary angioplasty (BPA) into clinical practice [5,6] and pulmonary vasodilators for inoperable cases [7,8,9,10] which represent approximately 50% of the total number of patients [11], has absolutely changed the therapeutic scenario.

PEA is still used less frequently in women despite CTEPH being equally prevalent both in women and men [12]. Nevertheless, specific comprehensive analyses of the patient phenotyping, clinical course, and prognosis in women with CTEPH are limited. Indeed, there is an unmet need for “real life” data to assess the impact of the introduction of the different therapeutic strategies on affected women. Thus, the present study sought to assess sex-based differences in clinical characteristics, treatment, and outcomes over time in a large and unselected cohort of CTEPH patients followed in a single reference national center for the study of pulmonary hypertension between 2007 and 2019.

## 2. Materials and Methods

### 2.1. Study Population

The REHAP (Registro Español de Hipertensión Pulmonar) has been previously described [13]. In short, this prospective registry started in 2007, documents cases of pulmonary arterial hypertension (PAH) and CTEPH in Spanish hospitals. For the present analysis, only patients with CTEPH followed up between 2007 and 2019 in Hospital Universitario 12 de Octubre, Madrid, Spain, were used. All records were adjudicated by a team of research physicians, using the definition of PAH and CTEPH specified by the existing European guidelines [14].

For each patient, the presence of cardiovascular (CV) risk factors was determined using data from REHAP, corresponding to the diagnostic period of CTEPH. The following risk factors were evaluated: diabetes, hypertension, dyslipidemia, smoking habits, and body mass index. Other significant comorbidities were also assessed: previous coronary artery disease, cancer history (active or considered resolved), history of prior PE episodes, and the previous diagnosis of a prothrombotic disorder (antiphospholipid syndrome and pathogenic variants in factor V Leiden or Prothrombin G20210A).

The follow-up time was defined as that from the diagnostic right heart catheterization (RHC) until the last follow-up completed in the REHAP registry. The vital status of the study patients at follow-up was also assessed.

### 2.2. Biomarkers, Functional Tests, and Hemodynamic Variables

The first value of plasma N-terminal pro-brain natriuretic peptide (NT-proBNP) was recorded when available if it was obtained at least six months around the diagnosis of the disease. The World Health Organization (WHO) functional class and the distance walked in the first six-minute walking test (6MWT) were abstracted from the REHAP registry. Hemodynamic variables were obtained from the diagnostic RHC.

### 2.3. Medical Therapies and Interventions

The type of treatment was decided on by the consensus of an expert panel in CTEPH comprised of at least one radiologist, one cardiac surgeon, one interventional cardiologist and one clinical cardiologist. Patients were divided according to the received treatment into three groups: PEA, BPA, and medical treatment only. Patients with BPA performed after PEA were categorized in the PEA group. Medication during the first evaluation in the referral center was recorded. Phosphodiesterase-5 inhibitors included sildenafil and tadalafil. Endothelin receptor antagonists included bosentan, macitentan, and ambrisentan. Prostanoids included epoprostenol and treprostinil. Riociguat was considered as a separate category (guanylyl cyclase stimulator), and this treatment was never administered with phosphodiesterase-5 inhibitors. The use of one, two, or three of these drugs at the same time was considered simple, double, or triple therapy, respectively.

The rationale for establishing two time periods (2007–2013 and 2014–2019) was done considering the addition of BPA as a treatment option for CTEPH patients during the first part of 2014.

### 2.4. Classification of Disease

The University of San Diego (USD)-California classification of disease was obtained from the surgical reports. Lesions involving at least one of the main pulmonary arteries were considered type I lesions. Lesions starting at the level of lobar branches were considered type II lesions. Lesions starting from segmental branches were catalogued as type III [15]. Similarly, disease localization in angiographic computed tomography (CT) of pulmonary arteries was categorized as described previously by our group [16]: proximal disease was defined as lesions in the proximal main, lobar, and proximal segmental arteries. Mid and distal segmental and subsegmental branches were considered peripheral diseases (Figure 1).

### 2.5. Outcomes

The primary outcome of interest was all-cause mortality in both sexes. Secondary outcomes included clinical and analytical differences, as well as differences in the localization of the disease and the received treatment in women and men over time.

### 2.6. Statistical Analysis

Categorical variables are reported as absolute and relative frequencies, and compared with Pearson’s or Fisher’s exact tests, as appropriate. Continuous variables are reported as means (Standard Deviation) or medians (Interquartile Range) and compared with t-tests or Mann–Whitney U tests, as appropriate. Kaplan-Meier survival curves were compared using the log-rank test. Cox proportional hazards modelling was used to assess the prognostic implications of the most relevant factors on all-cause mortality. Stratified analyses by sex were performed to assess separately in men and women the influence of each study variable on the estimated mortality. The statistical significance of the interaction term of each study variable with sex was also reported. A multivariable analysis with adjustment for sex, age, treatment, and time-period was also performed. Stata version 14.0 (StataCorp, College Station, TX, USA) and R studio v 4.0.3 (Boston, MA, USA) were used for data analysis.

## 3. Results

### 3.1. Study Population. A Global Gender Perspective in CTEPH

Our study population consisted of 453 patients with the diagnosis of CTEPH, of whom 252 (55.6%) were women and 201 (44.4%) men. Overall, 235 patients received PEA (51.9%), 91 BPA (20.1%), and 127 (28.0%) medical treatment without any other invasive intervention (Figure 2). There were no differences in the proportion of women between the two study periods (53.4% of women in the first era Vs 56.4% in the second era, *p* = 0.569).

Women were older and tended to have more cardiovascular risk factors. Nevertheless, more men were current or former smokers and had a lower value of FEV1. Women performed a shorter distance in the 6MWT compared with men (349 vs. 415 m, *p* < 0.001) and were significantly more likely to manifest a worse functional status (WHO functional class III or IV. 65.2% vs. 54.5%, *p* = 0.021) (Figure 3 and Appendix A). Women had significantly higher mean pulmonary resistances (9.8 vs. 8.5 WU) (Figure 4). Women were significantly more likely than men to be treated with triple vasodilator therapy (7.1% vs. 2.5%, *p* = 0.025), to receive diuretics (54.0% vs. 42.8%, *p* = 0.018), and to require oxygen supplementation (43.7% vs. 28.9%, *p* = 0.001).

### 3.2. Sex Differences in Treatment Groups and Disease Distribution

Patient characteristics at diagnosis stratified by treatment and sex are provided in Table 1. PEA was the treatment of choice for 61.7% of men and 44.1% of women (*p* < 0.001). A similar proportion of women and men received BPA, and more women received medical treatment only (Figure 3). There was no difference in the median age between both sexes in patients who underwent surgery. Nevertheless, women who ultimately received BPA or medical treatment exclusively were significantly older. The distance completed in the 6MWT was significantly shorter for women regardless of the received treatment. Notably, baseline PVR values were higher among women who received PEA in comparison with men, but mPAP, RAP, and CI were similar by sex in the three treatment groups (Appendix A). The main cause why BPA was carried out instead of PEA was the presence of a non-accessible disease for PEA in both sexes (96.6% for women and 87.9% for men, *p* = 0.277). On the other hand, the disproportion between high PVR and low thrombotic component obstruction was the main reason why exclusive medical therapy was prescribed (53.7% in women and 48.8% in men). Comorbidity was the second reason for indicating medical therapy only in a similar number of women and men (31.7% in women and 32.6% in men). The patient refusal was the third cause (13.4% in women and 18.6% in men, *p* = 0.76).

During the 2007–2013 era there were no differences in the surgical classification of disease between women and men (*p* = 0.169). Nevertheless, none of these patients had class III by the USD surgical classification. On the other hand, during the second period, surgical pieces among women were classified more frequently as class III (42.3% in women vs. 19.8% in men, *p* < 0.007). A non-significant higher proportion of distal disease was noticed in the diagnostic angiographic-CT among women (98.3% in women vs. 93.7% in men who underwent BPA, *p* = 0.519; 78.8% in women who received medical treatment exclusively vs. 65.9% in men, *p* = 0.124).

### 3.3. Therapeutic Trends

PEA procedures were carried out in 70 patients (59.3%) during the 2007–2013 period and in 165 (49.2%) during the 2014–2019 period. During both time periods, compared with males, females were more likely to receive medical therapy only, instead of any invasive therapy, with an important increase in the proportion of BPA in the second period in women and men (Figure 3). The 30-day mortality after PEA was 3.4%, similar for men and women (3.4% and 3.6%, respectively. *p* = 0.873). Only 1 woman out of the 91 patients treated with BPA died in the perioperative period. No men died during the same period. PVR decreased significantly both after surgery (9.2 ± 0.4 preoperatively decreased to 3.7 ± 0.2 WU after PEA, *p* < 0.001) and after the completion of BPA procedures (10.4 ± 0.5 pre-procedure decreased to 4.6 ± 0.2 WU, *p* < 0.001).

### 3.4. Prognosis

Over a median follow-up time of 50.7 months (interquartile range: 30.0–75.2 months), there were 42 deaths. Global survival at 1, 3, and 5 years was 97.1%, 94.0%, and 89.6%, respectively. The unadjusted risk of dying was similar in women and in men (Figure 5).

Only NT-proBNP levels and the use of targeted medical therapy were associated with higher mortality both in women and men. A lower mortality in the second time interval was observed only among women (HR 0.38, 0.16–0.87; *p* = 0.023). Equally, a longer distance walked in the 6MWT was associated with better survival exclusively in women. Age and dyslipidemia were associated with a higher risk of mortality only among women. On the contrary, BMI was associated with a higher risk of mortality only in men. Nevertheless, no differences in any of the studied variables were observed between men and women: the interaction term of each study variable with sex was not significant for any of the studied relationships.

After adjustment for sex, age, type of treatment, and time interval, the WHO functional class, the nt-proBNP value, and most of the hemodynamic variables (PVR, RAP, and mPAP) surged as risk factors for mortality. Moreover, the protective effect of the second interval of time observed in the unadjusted analysis was not confirmed (Table 2). For all the adjusted models, the variable sex was not significant.

## 4. Discussion

In this study we reviewed 453 consecutive CTEPH cases evaluated in a single referral center between 2007 and 2019. Despite treatment changes during these years, women continue to receive invasive procedures less frequently. However, crude mid-term mortality of women was similar when compared with men. Stratified and multivariable analyses showed that the variable sex was not an effect modifier, neither a confounding factor for mortality.

In our study, women were older, declared a poorer functional class and walked a shorter distance in the 6MWT. The 6MWT has demonstrated proven efficacy for prognostic evaluation of this disease [17]. Still, the distance walked could be skewed by musculoskeletal problems and is highly influenced by factors like muscle strength and the peripheral factor (artery-venous difference) [18]. Deconditioning and reduced respiratory muscle strength define the term physical frailty [19]. Both factors are common in CTEPH [20] and are especially relevant among elderly patients [21]. Likewise, comorbidity is an important feature linked with physical frailty [22]. Although some registries showed a similar age and a lower burden of cardiovascular risk factors in women [12], in our study a different profile of women with CTEPH is depicted, where women were clearly older and had a higher burden of cardiovascular risk factors. In our series, 6MWT had prognostic value, especially among women. Theoretically, this technique could be reflecting their higher physical frailty.

A small number of studies have evaluated sex differences in the treatment of CTEPH. Our study shows how treatment strategy differs depending on sex: PEA was developed more frequently in men, and medical therapy and oxygen supplementation were used more frequently in women. Data from the International Prospective Registry during the 2007–2009 period showed a higher proportion of women and a higher mean age among the non-operable population [11]. Not surprisingly, these results are repeated in recent registries, such as the US-CTEPH-R [23]. These inequalities probably reflect the fact that women more frequently have a non-accessible disease, as previously suggested by Barco et al. [12]. Although the latter was the first study which evaluated sex differences in CTEPH in Europe and Canada, important distinctions between our study and the former can be noted. That work describes the gender gap in treatment during the 2007–2012 period, but medical therapy and BPA were not considered. Plus, in our study women were significantly older and PVR was higher in them when compared with the female population of the previous work. The analysis of the development of contemporary therapies over time enhances the importance of cardiac surgeons and cardiovascular interventionist’s experience regarding the treatment of distal disease, especially for women. Notably, both treatment options and the time-period demonstrate its importance in the prognosis in the univariate analysis. Although more patients with distal disease could have been referred to our center since the BPA program started, the increase in the number of patients in the second time-period was balanced, corroborating the different disease profile in both sexes and the special meaning of the introduction of BPA for women. In our cohort, coronary artery disease was more prevalent among men, so coronary bypass surgery (CABG) was performed in more men at the same time as a PEA procedure, a similar finding when compared with the results of the previous studies [12]. However, a different distribution of cardiovascular risk factors is reported in our work: women had more diabetes, and dyslipidemia, probably reflecting their higher age. Finally, the patient’s refusal of surgery could have influenced the received treatment. Although previous work showed a female predominance among technically-operable-non-operated patients due to refusal of the procedure (40.9% of those did not undergo surgery for this reason. Of them, 63% were women) [24], rejection for surgery in technically-operable-non-operated patients in our series was slightly higher in males when compared with their female counterparts (10 out of 29-34.5%- among men vs. 12 out of 43-27.9%- among women, *p* = 0.552). Although women usually perceive a higher likelihood of negative outcomes and their probability of engaging in risky behaviors in terms of health is lower than men [25], our results do not confirm this tendency in the Spanish CTEPH population. Further work should be done to investigate if culturally assigned roles are associated with these results.

Disease distribution was also different in women. Hormone disparities have been proposed as a mechanism of the development of vascular pathology in the lungs in the case of PAH, and this could influence the higher prevalence of PAH among women [26]. Estrogen has also been related to disparities in the form of presentation of venous thromboembolism (VTE) in women with respect to men [27]. Even though PE is a more common form of presentation of VTE in women, they tend to be older when this first episode appears [28], and episodes in men tend to be more severe, and more repetitive than in women [29]. CTEPH is commonly associated with a past episode of PE. Subsequently, the different presentation of the acute episode of PE in women could influence the different profile of the chronic disease and the received treatment.

The number of PEA procedures steadily increased over time, as well as the complexity of the operated cases, considering that more distal cases were operated in the 2014–2019 period, especially among women. Similarly, BPA has increased robustly in both sexes since its introduction in 2014. Less women received any invasive treatment during both time intervals. The different patterns of CTEPH between both sexes could have influenced the received treatment, but many other factors could also be related. In recently published work by Hobohm et al. [30], the number of PEA also increased in Germany during the last 11 years, and in-hospital death decreased in parallel, suggesting that patient selection and institutional experience could be important determinants of outcomes. Regarding BPA, a contemporary experience of a Japanese referral center during the 2011–2019 period also showed that many more women were treated only with pulmonary vasodilators (66.7 of the total number of patients treated exclusively with medical therapy were women), and a higher number of women were treated with BPA (84.4% of the number of patients who only received BPA) when compared with the proportion of patients treated with PEA (56.2% of them women) or with the combination of PEA and BPA (64.0% of them women) [31]. Although the comparison with Japanese cohorts is difficult, considering women and BPA are overrepresented in these (women represent 76.2% in that cohort and BPA is carried out in 80.4% of that population), our results are comparable with other European and American registries in terms of short-term mortality (30-day mortality of 3.5%), operability rate (51.9%), and hemodynamical results (decrease of 5.5 and 5.8 WU after PEA and BPA, respectively) [11,12,23], reinforcing the usefulness of contemporary therapies among women in a western population.

Despite all the described differences, crude long-term mortality was similar between women and men. Some variables demonstrated a different prognostic value only for one sex group: age, dyslipidemia, 6MWT, and the first period were associated with higher mortality only among women and a higher BMI was associated with higher mortality only among men. Even though women in our cohort were older, had more comorbidity and worse hemodynamical values, and less frequently received the preferred treatment in current guidelines [14], mortality was similar in comparison with men. Bearing in mind the previous, mortality among women could have been further influenced by age, comorbidity, and the evolution of the currently available therapies when compared with men. It is possible that women benefitted especially from the introduction of the BPA program and the extension of PEA for distal cases, but unnoted confounding factors could also explain the observed differences. Worldwide women’s global mortality is lower [32]. In the setting of PAH, higher mortality in the elderly man has also been described [33]. Regardless, disproportionally higher mortality currently exists for young women in the setting of coronary artery disease, and they had also historically received guideline-based treatments less frequently [34,35]. In the case of CTEPH, evidence regarding sex differences in mortality is limited. In a recent Chinese CTEPH cohort, higher mortality was reported for men [36]. Likewise, in a pioneer Japanese cohort that evaluated sex differences in CTEPH, women were also approximately five years older than men and their disease was also more distal, postoperative mortality tended to be higher in men [37]. Moreover, no study showing data of gender differences in mortality among European or American countries using the currently available therapies were performed before. Our results may aid to generate new hypotheses for future studies concerning this disease.

## 5. Limitations

Our study has several limitations inherent to its observational nature. Approximately 80% of both women and men had a previous diagnosis of PE. Whether the different presentation of CTEPH is related to a different index episode of VTE is unclear, since previous episodes of DVP were not registered. Regarding physical frailty, one additional factor that could have been related is hypoxaemia. Nevertheless, in this registry we were not able to collect data concerning the baseline arterial oxygen content. Moreover, as stated throughout the manuscript, women required oxygen supplementation more frequently during follow-up, making this hypothesis plausible. Additionally, the study was carried out in a reference center for the study of PH, and more complex cases could be depicted here. Thus, the applicability of these results should therefore be viewed with caution in centers without the possibility of PEA, BPA, and medical therapy. Notwithstanding, our population reflects a real picture of the Spanish CTEPH population, since more than 60% of the total number of patients diagnosed are evaluated in this national referral center [38]. Finally, the number of patients is relatively low to make comparisons in some minoritarian groups. Even though causes of mortality were not studied, all-cause mortality could better reflect the associated comorbidity of the included cohort. Furthermore, the low number of events (mortality) did limit the number of candidate prognostic factors to be included in the models and prevented from developing and validating a predictive model. Even considering these limitations, the study provides new relevant information and demonstrates important differences in some of the most meaningful characteristics of this rare disease.

## 6. Conclusions

In a cohort of 453 consecutive patients with chronic thromboembolic pulmonary hypertension evaluated from 2007 to 2019, there were several sex-specific differences. Women were older, complained of more symptoms, and had higher pulmonary vascular resistances before any intervention was carried out, suggesting a more established pulmonary vascular disease. The distribution of the disease was also different. Supporting the previous, more complex cases were operated over time, particularly among women. Balloon pulmonary angioplasty vigorously increased during the 2014–2019 interval, equally in men and women. Nevertheless, women were treated invasively less frequently and needed more medical treatment in both periods. Despite the previous differences, mortality was comparable in men and women. The most recent time period was associated with a higher survival only among women, suggesting that the improvement of pulmonary endarterectomy and the introduction of balloon pulmonary angioplasty has been more determinant for women. Further well-designed investigations studying sex specific differences in CTEPH are needed to confirm our results.

## Figures and Tables

**Figure 1 jcm-10-04251-f001:**
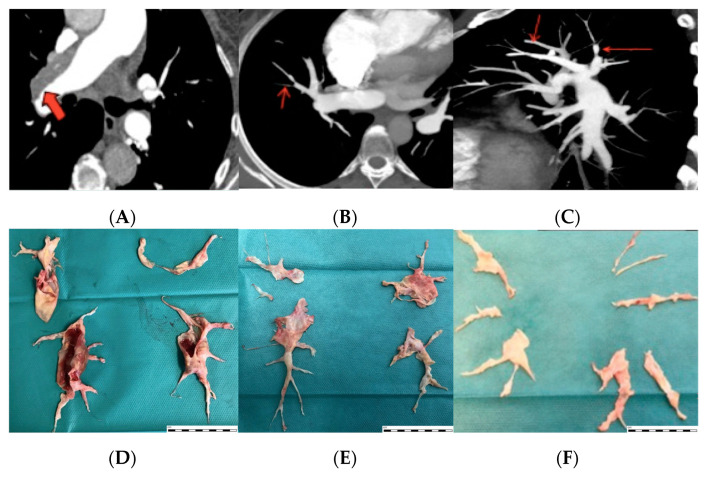
A comparison of angiographic CT images and surgical pieces of pulmonary endarterectomy. In panel (**A**) there is an image of proximal thromboembolic disease in the right pulmonary artery (red thick arrow), more frequent among men. In panels (**B**,**C**) some images of distal thromboembolic disease are pointed out (red thin arrows), a disease distribution more frequent in women. Panels (**D**–**F**) show pieces of PEA classified as level I, II, and III after surgery, respectively.

**Figure 2 jcm-10-04251-f002:**
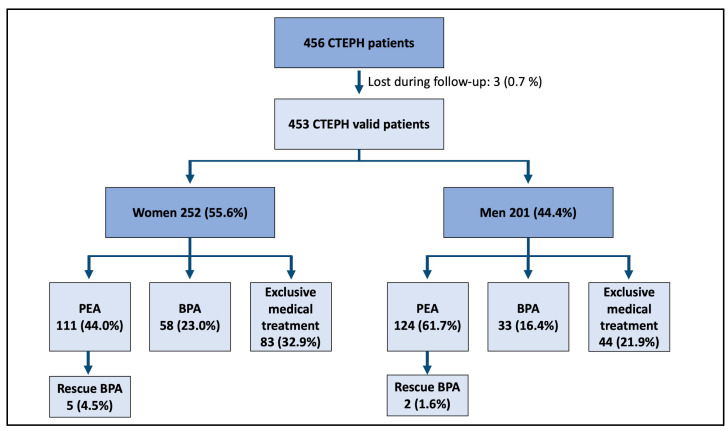
Flow chart of included patients. CTEPH (chronic thromboembolic pulmonary hypertension), PEA (pulmonary endarterectomy), BPA (balloon pulmonary angioplasty).

**Figure 3 jcm-10-04251-f003:**
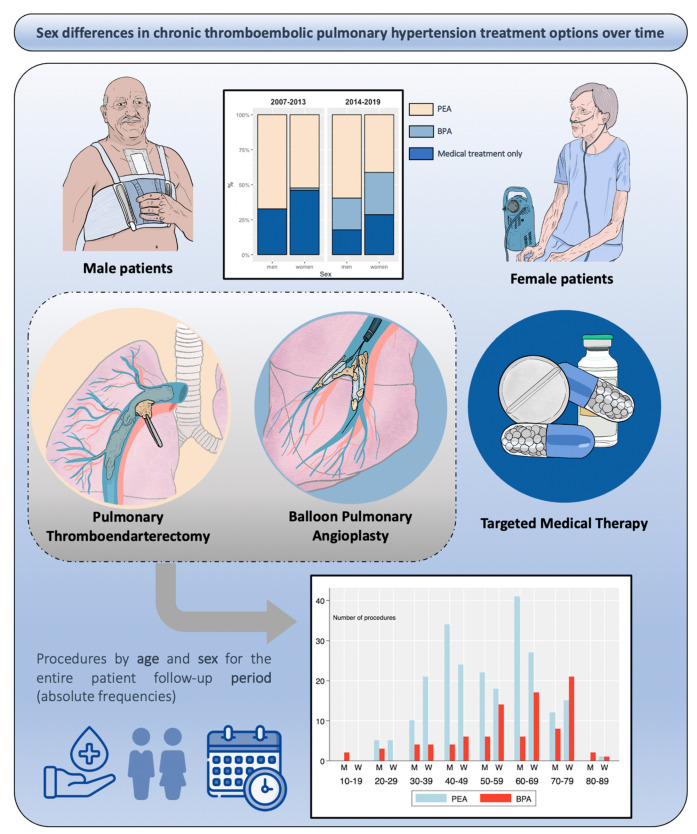
Treatment options over time (2007–2013, *n* = 118; 2014–2019, *n* = 335) by sex (relative frequencies). Procedures by age and sex for the entire patient follow-up period (absolute frequencies).

**Figure 4 jcm-10-04251-f004:**
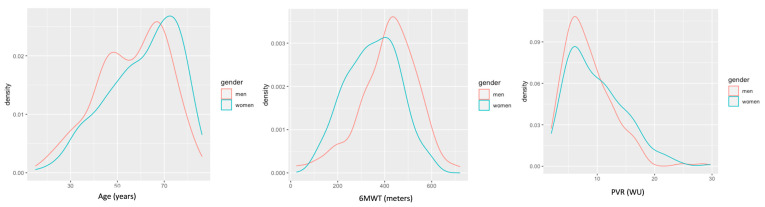
The distribution of relevant baseline characteristics in women and men. In the left panel, two peaks of a younger age can be seen for men. In the central panel, there is a homogeneous and higher distribution of the walked distance for men. In the right panel, there is also a more homogeneous and lower distribution of the PVR for men in comparison with a more heterogeneous and higher distribution of PVR among women. 6MWT (six-minute walking test), PVR (pulmonary vascular resistance), WU (Wood Units).

**Figure 5 jcm-10-04251-f005:**
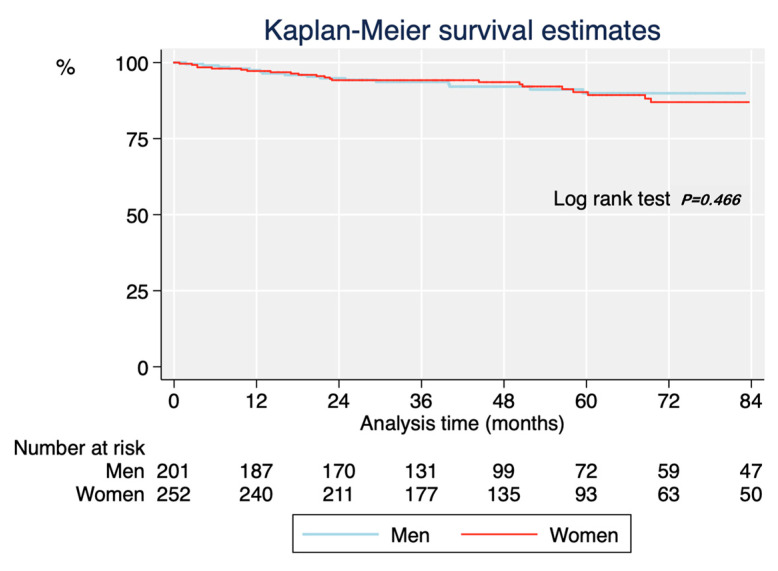
Crude Kaplan-Meier survival curves by sex.

**Table 1 jcm-10-04251-t001:** CTEPH patient characteristics at diagnosis by type of procedure and sex.

CTEPH Patient Characteristics at Diagnosis by Type of Procedure and Sex (*n* = 453).												
	PEA				BPA				MT			
	*n* (%N)	Women	Men	*p* Value	*n* (%N)	Women	Men	*p* Value	*n* (%N)	Women	Men	*p* Value
Number of patients—(*n*/%)	235 (100.0)	111 (44.1)	124 (61.7)	<0.001	91 (100.0)	58 (41.1)	33 (42.9)	0.805	127 (100.0)	83 (32.9)	44 (21.9)	0.009
Age—years (median—IQr)	235 (100.0)	53.5 (41.2–66.2)	55.8 (45.5–65.7)	0.672	91 (100.0)	66.2 (56.3–74.5)	57.4 (40.6–72.0)	0.033	127 (100.0)	71.7 (59.6–71.7)	66.0 (50.7–73.4)	0.013
Arterial hypertension (*n*/%)	235 (100.0)	36 (32.4)	48 (38.7)	0.316	91 (100.0)	31 (53.5)	10 (30.3)	0.033	127 (100.0)	40 (48.2)	19 (43.2)	0.590
Diabetes (*n*/%)	235 (100.0)	12 (10.8)	9 (7.3)	0.341	91 (100.0)	10 (17.2)	1 (3.0)	0.046	127 (100.0)	13 (15.7)	5 (11.4)	0.509
Dyslipidemia (*n*/%)	235 (100.0)	25 (22.5)	25 (20.2)	0.659	91 (100.0)	14 (24.1)	7 (21.2)	0.750	127 (100.0)	36 (43.4)	10 (22.7)	0.021
BMI (median*—*IQr)	231 (98.3)	29.4 (23.4–31.4)	27.2 (24.4–30.1)	0.417	74 (81.3)	28.2 (26.0–32.0)	26.8 (23.5–30.5)	0.114	102 (80.3)	28.0 (24.2–31.0)	29.0 (26.0–31.0)	0.240
Coronary artery disease (*n*/%)	235 (100.0)	5 (4.5)	9 (7.3)	0.373	91 (100.0)	1 (1.7)	2 (6.1)	0.265	127 (100.0)	4 (4.8)	4 (9.1)	0.346
Functional class—NYHA (*n*/%)III-IV	218 (92.8)	77 (74.8)	77 (67.0)	0.207	90 (98.9)	37 (63.8)	15 (46.9)	0.120	127 (100.0)	50 (60.2)	17 (38.6)	0.020
Six-minute walking test—meters (mean ± SD)	159 (67.7)	366.2 ± 12.7	425.6 ± 14.6	0.003	87 (95.6)	335.4 ± 15.2	423.3 ± 20.5	0.001	117 (92.1)	340.9 ± 12.3	388.3 ± 19.3	0.034
Nt-proBNP—mg/dL (median—IQr)	149 (63.4)	643.5 (289.0–2012.0)	478.0 (86.0–1912.0)	0.275	89 (97.8)	633.0 (175.0–2079.0)	702.5 (244.5–1369.5)	0.949	110 (86.6)	290.5 (126.5–1823.5)	262.0 (105.0–1087.0)	0.375
FEV1*—*% predicted (mean ± SD)	137 (58.3)	82.5 ± 1.7	78.7 ± 1.8	0.126	72 (79.1)	89.6 ± 2.4	80.7 ± 3.8	0.044	105 (82.7)	91.5 ± 2.5	84.4 ± 2.9	0.074
FVC*—*% predicted (mean ± SD)	133 (56.6)	89.0 (73.0–99.0)	86.5 (76.0–96.0)	0.693	81 (89.0)	95.0 (74.0–105.0)	84.2 (74.0–98.0)	0.206	103 (81.1)	93.0 (80.0–110.0)	90.7 (83.0–100.5)	0.349
PE history (*n*/%)	235 (100.0)	91 (82.0)	102 (82.3)	0.956	91 (100.0)	42 (72.4)	21 (63.6)	0.383	127 (100.0)	65 (78.3)	36 (81.8)	0.641
Hypercoagulability (*n*/%)	182 (77.4)	57 (60.6)	53 (60.2)	0.955	86 (94.5)	18 (32.7)	11 (35.5)	0.795	118 (92.9)	33 (41.8)	16 (41.0)	0.938

BMI (body mass index); BPA (balloon pulmonary angioplasty); DLCO (diffusing capacity of the lung for carbon monoxide); FEV1 (forced expiratory volume in one second); FVC (forced vital capacity); MT (medical treatment); PE (pulmonary embolism); PEA (pulmonary endarterectomy).

**Table 2 jcm-10-04251-t002:** Mortality Cox regression analysis.

Unadjusted, Stratified by Sex and Adjusted Univariate Cox Regression Analysis for Mortality					
	Crude HR (CI 95%)	*p* Value	Stratified by Sex (CI 95%)	*p* Value	Adjusted * (CI 95%)
Sex (women vs. men)	1.26 (0.68–2.35)	0.467			0.98 (0.52–1.87)
Age (Per 10-year increment)	1.43 (1.13–1.81)	0.003	Men 1.33 (0.91–1.93) Women 1.52 (1.12–2.06)	0.138 0.007	1.02 (0.99–1.04)
BMI—Kg/m^2^ (By increments of 10 Kg/m^2^)	1.03 (0.96–1.10)	0.458	Men: 1.14 (1.02–1.27) Women: 0.97 (0.88–1.06)	0.024 0.491	1.01 (0.94–1.09)
Systemic hypertension	1.75 (0.96–3.21)	0.070	Men: 1.34 (0.50–3.59) Women: 2.13 (0.97–4.64)	0.566 0.058	1.38 (0.73–2.59)
Diabetes mellitus	1.84 (0.82–4.17)	0.078	Men: 0.77 (0.10–5.84) Women: 2.50 (0.99–6.29)	0.802 0.052	1.59 (0.70–3.63)
Dyslipidemia	1.18 (0.60–2.30	0.637	Men: 0.1 (0.1-inf) Women: 2.24 (1.03–4.87)	1.000 0.041	0.80 (0.40–1.62)
Smoking habits	1.00 (0.53–1.90)	0.995	Men: 0.65 (0.24–1.80) Women: 1.46 (0.63–3.36)	0.410 0.376	1.27 (0.64–2.52)
Cancer history	1.27 (0.56–2.86)	0.566	Men: 1.43 (0.41–5.02) Women: 1.15 (0.39–3.33)	0.576 0.802	1.16 (0.51–2.63)
Coronary artery disease	1.69 (0.60–4.74)	0.318	Men: 1.66 (0.38–7.30) Women: 1.70 (0.40–7.21)	0.503 0.473	1.46 (0.52–4.13)
6MWT—meters (by increments of 30 m)	0.74 (0.62–0.89)	0.002	Men: 0.86 (0.60–1.23) Women: 0.64 (0.50–0.83)	0.417 0.001	0.70 (0.56–0.87)
NtproBNP—mg/dL (by increments of 1000)	1.23 (1.14–1.23)	<0.001	Men: 1.21 (1.04–1.42) Women: 1.21 (1.11–1.32)	0.017 <0.001	1.23 (1.14–1.33)
WHO III-IV (reference WHO FC I-II)	1.69 (0.83–3.45)	0.149	Men: 1.24 (0.42–3.64) Women: 2.18 (0.82–5.79)	0.310 0.117	2.55 (1.22–5.36)
PVR—WU (by increments of 1 mmHg)	1.04 (0.99–1.11)	0.144	Men: 1.01 (0.90–1.13) Women: 1.05 (0.98–1.13)	0.907 0.160	1.08 (1.02–1.14)
CI—L/min/m^2^ (by increments of 1 l/m/m^2^)	0.75 (0.46–1.25)	0.271	Men: 0.93 (0.40–2.16) Women: 0.67 (0.36–1.26)	0.864 0.217	0.62 (0.37–1.03)
mPAP—mmHg (by increments of 10 mmHg)	1.19 (0.91–1.55)	0.202	Men: 1.32 (0.85–2.04) Women:1.12 (0.80–1.57)	0.215 0.496	1.46 (1.11–1.91)
RAP—mmHg (by increments of 1 mmHg)	1.06 (1.01–1.11)	0.044	Men: 1.05 (0.96–1.14) Women: 1.05 (0.99–1.13)	0.266 0.121	1.09 (1.03–1.15)
Type of treatment received (in reference to PEA)—BPA —Medical therapy	1.20 (0.37–3.82) 6.23 (3.02–12.85)	0.763 <0.001	Men: 1.67 (0.32–8.65) Women: 0.90 (0.17–4.65) Men: 5.53 (1.85–16.54) Women: 6.66 (2.47–17.93)	0.542 0.900 0.002 <0.001	1.39 (0.41–4.71) 4.91 (2.22–10.85)
Period of time (2014–2019 Vs 2007–2013)	0.48 (0.25–0.91)	0.024	Men: 0.65 (0.24–1.77) Women: 0.38 (0.16–0.87)	0.398 0.023	0.58 (0.29–1.16)

Crude, stratified by sex and * adjusted for sex, age, treatment, and period of time. BMI (body mass index); BPA (balloon pulmonary angioplasty); CI (cardiac index); mPAP (mean pulmonary artery pressure); PEA (pulmonary artery endarterectomy); PVR (pulmonary vascular resistance); RAP (right atrium pressure); WHO (World Health Organization); WU (Wood Units); 6MWT (six-minute walking test).

## Data Availability

All the relevant data is included within the manuscript and Appendix A.

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
