# Peer review of "Sex Differences in Chronic Thromboembolic Pulmonary Hypertension. Treatment Options over Time in a National Referral Center"

_jcm, 2021, doi:10.3390/jcm10184251_

Round 1

Reviewer 1 Report

This is an interesting study showing gender difference in CTEPH after BPA was introduced, although several studies already showed gender difference in CTEPH.

Major

Do you have any blood gas data, because Japanese study showed worse PaO2 in women.

Hypoxemia might have effects on worse functional class and 6MWT.

Women outnumbered man.

Is there any difference in female to male ratio between period of time (2014-2019 vs. 2007-2013)? Because more distal type patients might be often diagnosed after BPA started.

A Japanese study showed female patients had less DVT and less acute embolic episodes?

How about in this study?

I think that first part of Materials and methods should be excluded.

Minor

“women had 325 more diabetes, systemic hypertension”

There was no difference in hypertension between male and femal in this study.

Is it right?

“by Barco et al [12]. Although that last study was the first which evaluated sex differences in CTEPH”

I do not think so.

Shigeta showed gender difference in CTEPH in 2008.

Reviewer 2 Report

Sex Differences in CTEPH. Treatment Options Over Time in a National Referral Center

This is a register-based analysis of data including 453 patients with CTEPH with respect to outcome in two time periods (2007-14 and 2015-19) with special focus on sex differences. The two time frames are distinct by the introduction of balloon pulmonary angioplasty (BPA) from 2015 on. Mortality was generally low (11.4% after 5 years) and did not differ between male and female patients, although female patients were generally older and were less frequently treated by invasive techniques (pulmonary endarteriectomy (PEA) or BPA). Interestingly, a lower mortality in the second time interval was only observed for women.

Comment

This is an interesting analysis of the important subgroup of patients with pulmonary hypertension who suffer from CTEPH. The results are reassuring with a low mortality (when compared e.g. with patients who suffer from critical limb ischemia). The reason why mortality did not improve after the introduction of BPA, however, should be better discussed. A registry has the draw-back in that it is quite difficult to separate prevalent from incident patients. Could it be that in the second time interval there were more incident patients (who have a worse prognosis) so that the lack of improvement in mortality could be explained by this characteristic? Also, it is difficult to understand why women had better outcomes in the second time interval although they had less invasive treatments. There must be unnoted confounding factors perhaps among male patients which conferred a lack of improvement in the second time frame among males.

All in all, the data are of interest and are reassuring, but the explanation of the findings may be incomplete (perhaps also due to the inherent problematics with registries). The authors should acknowledge the fact that the interpretation of the findings should not be overstressed.

Minor point

Line 62-76 have to be cancelled.

Round 2

Reviewer 1 Report

There are no further comments.
